# The incidence and predictors of peripheral arterial disease among type 2 diabetes mellitus patients at Felege Hiwot Comprehensive Specialized Hospital, Northwest Ethiopia, 2023: A retrospective follow-up study

**Dessalew Abelneh Woleli[1], Gebiyaw Wudie Tsegaye[2], Taye Abuhay[2], Abyot Terefe Teshome[1], Gebrie Getu Alemu[3]***

1 Department of Internal Medicine, Felege-Hiwot Comprehensive Specialized Hospital, Bahir Dar, Ethiopia, 2 Department of Epidemiology and Biostatistics, School of Public Health, College of Medicine and Health Sciences, Bahir Dar University, Bahir Dar, Ethiopia, 3 Department of Epidemiology and Biostatistics, Institute of Public Health, College of Medicine and Health Sciences, University of Gondar, Gondar, Ethiopia

* gebryegetu27@gmail.com

## Abstract

### Background

Peripheral arterial disease is a blockage or narrowing of arteries transporting blood from the heart to the legs and feet. Type 2 diabetes mellitus is the second most common risk factor for peripheral arterial disease. This condition is associated with patient mortality and morbidity. However, there is limited evidence on the time to develop peripheral arterial disease, its incidence, and its predictors in the Amhara region, Ethiopia.

### Objective

To assess the incidence and predictors of peripheral arterial disease among adult type 2 diabetes mellitus patients at Felege Hiwot Comprehensive Specialized Hospital, Northwest Ethiopia, 2023.

### Methods

A retrospective follow-up study was conducted using data from 552 type 2 diabetes mellitus patients, selected by simple random sampling from January 2006 to December 2021. Data from patient charts and follow-up forms were extracted using a standardized tool, entered into Epi-Data version 3.1, and analyzed in STATA version 14. Kaplan-Meier curves, life table, and Cox regression model were used to assess survival times, survival probabilities, and identify independent predictors of peripheral arterial disease, respectively.

**Data availability statement:** All relevant data are within the manuscript.

**Funding:** The author(s) received no specific funding for this work.

**Competing interests:** The authors have no competing interest.

**Abbreviations:** ABI, Ankle Brachial Index; ADA, American Diabetes Association; AHR, Adjusted Hazard Ratio; CAD, Coronary Artery Disease; CI, Confidence Interval; CVD, Cardiovascular Disease; DB, Diastolic Blood Pressure; DM, Diabetes Mellitus; GBD, Global Burden of Disease; GFR, Glomerular Filtration Rate; HDL, High Density Lipoprotein; HTN, Hypertension; IDF, International Diabetes Federation; LDL, Low Density Lipoprotein; MRC, Medical Referral Clinic; PAD, Peripheral Arterial Disease; SBP, Systolic Blood Pressure; SSA, Sub Saharan Africa; TC, Total Cholesterol; TG, Triglyceride; WHO, World Health Organization.

## Results

The final analysis included 552 records. The median time for peripheral arterial disease was 13.5 years (95% CI: 11, 14.5). Being female (Adjusted Hazard Ratio (AHR) = 2.18, 95% CI: 1.36, 3.51), age above 65 years (AHR = 1.66, 95% CI: 1.06, 2.61), and fasting blood sugar of over 140 mg/dl (AHR = 3.34, 95% CI: 1.62, 6.90) were predictors for time to peripheral arterial disease in type 2 diabetes mellitus patients. We observed an incidence rate of 29 cases of peripheral arterial disease per 1000 person-years of observation (95% CI: 24–36).

## Conclusion and Recommendation

Most peripheral arterial disease occurs after 10 years of diagnosis of type 2 diabetes mellitus. The incidence rate of peripheral arterial disease was high. Female sex, older age, and elevated baseline fasting blood sugar predicted the time until peripheral arterial disease developed. Therefore, clinicians and other stakeholders shall emphasize those types of type 2 diabetes mellitus patients who are female, older age, and with raised fasting blood sugar.

## Introduction

Peripheral Arterial Disease (PAD) is the blockage or narrowing of arteries that transport blood from the heart to the legs and feet. It results commonly from atherosclerosis, the buildup of fatty plaques in the arteries [1]. Clinicians diagnose PAD based on clinical evidence such as intermittent claudication, absent or decreased peripheral arterial pulses, and Doppler ultrasound [2].

The commonly identified risk factors for PAD include hypertension (HTN), hypercholesterolemia, smoking, and a previous history of cardiovascular diseases [3]. Of which, diabetes is the 2nd most common risk factor following smoking [4]. Peripheral arterial disease is 3–4 times more prevalent and severe in diabetic patients as compared to non-diabetic ones [5,6]. Moreover, patients with PAD who have type 2 diabetes are more likely to develop cardiovascular illnesses, die from those illnesses, and have a higher likelihood of lower extremity amputation [3,7,8].

Peripheral arterial disease, along with other associated cardiovascular diseases, has put a threat on the globe. This impact is no longer limited to the developed nations but has streamed to the developing countries too. Diabetes mellitus (DM) has made human life susceptible to this morbidity; mainly, type 2 is responsible [9,10].

According to a comprehensive study conducted in 21 African nations, PAD prevalence ranged from 30.2 to 31.6% in individuals with diabetes. Therefore, the continent faces substantial public health and socioeconomic issues because of the pandemic surge in diabetes and its complications. The severity and magnitude are getting worse in this part of the world [11,12].

Among diabetic patients in Ethiopia, the prevalence of PAD and associated micro- and macrovascular consequences has been rising. The longer illness duration, the presence of additional comorbidities, and aging were all factors contributing to diabetes complications. Ethiopia is the first among the top five countries in Africa for the number of people with diabetes [13]. Some studies have estimated the median times for onset of PAD in type 2 diabetes mellitus (T2DM) patients, ranging from 3 years in New Zealand [14] to 26.97 years in Ecuador [15].

Early detection and treatment of PAD in diabetic patients are crucial for risk factor modification, lowering the prevalence and progression, and improving the outcome of the condition [16]. Enhancing quality of life, avoiding heart attacks, and lowering the risk of long-term incapacity and other issues are connected to it [17]. However, most T2DM patients didn't report intermittent claudication because of the asymptomatic nature of the disease masked by peripheral neuropathy. These asymptomatic natures, lack of awareness, and underutilization of screening tools made PAD underestimated and untreated [18,19].

Diabetes mellitus and other chronic diseases are on the alarming rise, putting the double burden of disease in Ethiopia. Despite this, there are limited studies on PAD and other vascular complications of type 2 diabetes concerned with the survival time of type 2 DM patients from PAD incidence and its predictors. So far, in Ethiopia and sub-Saharan Africa, studies conducted on macrovascular and other chronic complications of diabetes are not adequate. Even the scant epidemiological studies on PAD from DM in Africa, including Ethiopia, have only used cross-sectional studies to estimate prevalence. Knowing the incidence and the approximate time when PAD will occur after a patient gets type 2 diabetes is most important, as it answers when to start prophylaxis for this complication and minimizes the occurrence rate. This study will be important for policymakers, clinicians, and researchers to reduce PAD-associated morbidity and mortality among DM patients. Therefore, this study was aimed at filling the gap by determining the time to develop PAD, the incidence, and its predictors among type 2 DM patients at Felege Hiwot Comprehensive Specialized Hospital, Northwest Ethiopia.

## Methods and Materials

### Study design, area, and period

An institution-based retrospective follow-up study was conducted at Felege Hiwot Comprehensive Specialized Hospital (FHCSH), Northwest Ethiopia, from January 2006 to December 2021. It is one of the earliest comprehensive specialized hospitals in the Amara regional state, located in Bahir Dar town, 565 km from Addis Ababa, the capital city of Ethiopia. It was established in 1969 as a district hospital and was upgraded to a referral hospital in 2002 and to a comprehensive specialized hospital in 2017. Currently, it serves to more than 7 million people in the region. There are 2561 type 2 DM patients currently on follow-up at this hospital. Diabetic and other patients with chronic diseases are being followed by a team comprising internists, one family medicine specialist, and nurses.

### Source population

All type 2 diabetic patients aged 14 years and older receiving follow-up care at FHCSH.

### Study population

All newly diagnosed patients with type 2 diabetes mellitus (aged over 14 years) at FHCSH from January 2006 to December 2021.

### Inclusion criteria

All type 2 diabetic patients aged over above 14 years who were enrolled at FHCSH between January 1, 2006, and December 31, 2021.

### Exclusion criteria

Patients diagnosed with both diabetes and PAD simultaneously.

## Sample size determination

The necessary sample size was calculated using a two population proportion formula using Epi Info 7, with the following parameters: a 95% CI, a 5% margin of error, 85% power, and an AHR of significant factors based on a previous study [5]. The final sample size consisted of 552 participants. The study units were then selected through computer-generated random sampling methods using their medical registration numbers.

## Sampling technique and procedure

A simple random sampling method was employed to select the study units from the source population. A sampling frame was created using patients' medical registration numbers, which were gathered from the Health Management Information System (HMIS) registration book. These registration numbers were entered into Microsoft Excel and subsequently transferred to SPSS version 23. SPSS was utilized to generate random numbers corresponding to the patients' medical registration numbers, and the necessary folders were selected from these numbers.

## Data collection tool and procedure

Data were accessed for research purposes from December 1, 2022 to January 31, 2023. Extraction was performed through chart reviews using a structured checklist adapted from various literature sources and aligned with health facility formats. The tool was designed to gather socio-demographic, clinical, and laboratory-related information essential for assessment. Study participants were enrolled between January 1, 2006, and December 31, 2021, with follow-up conducted from their enrollment until the occurrence of the event. Initially, medical charts of eligible adult type 2 diabetes were collected from the card room using the patient's medical registration numbers. Basic socio-demographic, clinical, and biochemical variables were then extracted from those selected charts. Laboratory test results and clinical findings recorded at the start of follow-up were considered baseline values. Data collection was carried out by two BSc nurses under the supervision of a general practitioner from December 1, 2022, to January 31, 2023. Finally, the principal investigator verified the consistency between records and collected data by randomly reviewing of some the previously extracted medical records.

## Study variables

**Dependent variable.** Incidence of peripheral arterial disease.

## Independent variables

**Socio-demographic Factors:** Patients age, sex and place of residence.
**Baseline clinical factors:** Treatment type, history of hypertension, presence of retinopathy, peripheral neuropathy, family history of hypertension, coronary artery disease, family history of diabetes mellitus, systolic blood pressure, diastolic blood pressure, and pulse pressure.
**Baseline biochemical factors:** proteinuria, low-density lipoprotein (LDL), high-density lipoprotein (HDL), triglyceride (TG), total cholesterol (Tc), serum creatinine (Cr), hemoglobin A1C (HgbA1C), fasting blood sugar (FBS)

## Operational definitions

**Censored:** Was deemed lost to follow-up, died, transferred before experiencing the event, or remained event-free by the conclusion of study period.

**Event:** The occurrence of PAD in patients with type 2 DM during the follow-up.

**Incidence rate of the event:** The frequency of PAD occurrence during the follow-up period relative to the total person-years of observation.

**Time to PAD:** The duration between the diagnosis of newly cases of type 2 diabetes mellitus and the first occurrence of PAD measured in years of observation.

**Retinopathy:** A range of retinal abnormalities identified through ophthalmoscopic examination [20].

**Peripheral neuropathy:** An abnormality of the peripheral nerve characterized by limb pain and irregularities in nerve conduction tests [21].

**Peripheral Arterial Disease:** A document diagnosis in the patient's medical record that includes associated signs and symptoms or results from a Doppler ultrasound examination.

**Data quality assurance:** Supervisors and data collectors underwent a one day training and orientation session covering the study's objective, data collection procedure, and data extraction checklist. Throughout data collection, the supervisor conducted daily monitoring. The principal investigator verified the consistency and completeness of the data on a daily basis. Before data entry, the collected data were reviewed for completeness. To minimize errors, Epi-Data software was utilized for data entry.

## Data processing and analysis

A descriptive analysis was conducted to illustrate the proportion of each study variable. The Kaplan-Meier survival curve was employed to estimate survival probabilities, while log-rank tests were utilized to compare the survival curves. The probabilities of survival for each event within time intervals, as well as the cumulative probability of survival for subsequent time interval were estimated using the life table. A bi-variable Cox proportional hazard model was applied to identify candidate variables for multivariable Cox regression. The multivariable Cox proportional hazard model was then used to determine factors associated with the incidence of PAD. Variables with a p-value less than 0.25 in the bi-variable analysis were included in multivariable Cox regression analysis. Ultimately, the AHR with a 95% CI was calculated, and variables with a p-value less than 0.05 in the multivariable analysis were deemed significant predictors of PAD. Additionally, the outcome status was determined by dividing the total number of occurrences during the follow-up period by the total number of observations. Incidence density was measured with person-years of observation. The goodness of fit for the Cox regression model was evaluated through the Cox-Snell residual plot and the Schoenfeld residual global proportional hazard test. A nearly constant smoother line on the Schoenfeld residual plot, and a P-value of greater than 0.05 from the Schoenfeld residual proportional hazard (PH) test, indicated whether the proportional hazard assumption was upheld or violated.

## Ethical consideration

Ethical approval for this study was obtained from the Bahir Dar University College of Medicine and Health Sciences Ethical Review Board (IRB) under protocol number 597/2022. Since the study involved reviewing medical records, individual patients were not exposed to harm, and an official cooperation letter was obtained from Bahir Dar University to Felege Hiwot Comprehensive Specialized Hospital. Permission was granted by the hospital manager and the leader of medical ward unit. According to the research review committee of Bahir Dar University, written consent was not necessary as confidentiality and anonymity were rigorously upheld, and this requirement was waived by the IRB. To ensure confidentiality, patient names and other identifying details were excluded from the data extraction format;

data access was restricted solely to the principal investigator. Moreover, the data were not disclosed to anyone other than the principal investigator. Anonymity was maintained throughout the research process, and privacy measures were implemented to protect participants' during the study and its publication. This study adhered to the principles outlined in the Declaration of Helsinki.

## Results

### Socio-demographic characteristics of participants

A total of 552 medical charts of adult patients with type 2 DM, enrolled between January 2006 and December 2021, were reviewed and included in the final analysis. More than half (56.7%) of the study participants were females. More than three-quarters (81.34%) of the study participants were urban residents. About three-fourths (75.72%) of the study participants were with an age above 65 years (older age) (**Table 1**).

### Baseline and follow up clinical, and laboratory values

Approximately three-quarters of the study participants (75.72%) had reported having a family history of DM, while nearly half (47.46%) had a family history of HTN. Additionally, 111 participants (20.11%) were hypertensive themselves. Approximately one-third of the study participants had specific health conditions: 32.97% suffered from coronary artery disease, 30.98% experienced diabetic retinopathy, and 33.70% had peripheral neuropathy, respectively. Baseline measurements of blood pressure showed that 57 (10.33%) of the participants had a systolic blood pressure (SBP) > 140 mmHg, and 45 (8.15%) of the participants had a diastolic blood pressure (DBP) > 90 mmHg. From the lipid profile values at baseline, 122 (22.10%) had low ($\leq$ 40 mg/dl), 309 (55.78%) had moderate (40–60 mg/dl), and 121 (29.92%) had high (> 60 mg/dl) HDL levels. Regarding total cholesterol TC ($\geq$ 200 mg/dl) and TG (>150 mg/dl) were more than half in number as compared to those having optimal values, 380 (63.30%) and 474 (88.04%), respectively. Most of the participants, 475 (86.05%), had an optimal level of LDL $\leq$ 130 mg/dl. In the other laboratory results, more than half (71.74%) of patients had FBS > 140 mg/dl. Most of the study participants (81.34%) had HbA1C > 7%. Regarding the type of treatment, more than half (344, or 62.32%) were treated with non-insulin drugs, 162 (29.35%) were on mixed-drug treatment, and 41 patients (8.3%) were treated with insulin (**Table 2**).

### Incidence and survival from PAD in type 2 DM

The study participants were followed for a minimum of 1 year to a maximum of 15 years, which provides 3693 person-years of observation (PYO), and the overall incidence rate of

**Table 1. Socio-demographic characteristics of type 2 DM patients on follow-up at Felege Hiwot Comprehensive Specialized Hospital, Northwest Ethiopia, 2023 (n = 552).**

| Variables | Category of variables | Outcome Status | | Total (%) |
|---|---|---|---|---|
| | | PAD (Count, %) | Censored (Count, %) | |
| Sex of the patient | Male | 25 (10.46) | 214 (89.54) | 239 (43.3) |
| | Female | 84 (26.84) | 229 (73.16) | 313 (56.7) |
| Age of the patient | ≤65 | 45 (10.77) | 373 (89.23) | 418 (75.72) |
| | >65 | 64 (47.76) | 70 (52.24) | 134 (24.28) |
| Residence | Rural | 9 (8.74) | 94 (91.26) | 103 (18.66) |
| | Urban | 100 (22.27) | 349 (77.73) | 449 (81.34) |

**Table 2. Baseline and Follow up clinical and laboratory related characteristics of type 2 DM patients on follow-up at Felege Hiwot Comprehensive Specialized Hospital, Northwest Ethiopia, 2023(n = 552).**

| Variables | Category of variables | Outcome Status | | Total (%) |
|---|---|---|---|---|
| | | PAD (Count, %) | Censored (Count, %) | |
| Family history of DM | Yes | 24(17.91) | 110(82.09) | 134(24.28) |
| | No | 85(20.33) | 333(79.67) | 418(75.72) |
| Family History of HTN | Yes | 59(22.52) | 203(77.48) | 262(47.46) |
| | No | 50(17.24) | 240(82.76) | 290(52.54) |
| Type of treatment | Insulin | 6(13.04) | 40(86.96) | 46(8.30) |
| | Non-insulin drug | 79(22.97) | 265(77.03) | 344(62.32) |
| | Mixed | 29(17.90) | 133(82.01) | 162(29.35) |
| Hypertension | Yes | 43(37.84) | 68(61.26) | 111(20.11) |
| | No | 66(14.97) | 375(85.03) | 441(79.89) |
| CAD | Yes | 49(26.92) | 133(73.08) | 182(32.97) |
| | No | 60(16.22) | 310(83.78) | 370(67.03) |
| Retinopathy | Yes | 52(30.77) | 119(69.23) | 171(30.98) |
| | No | 57(14.88) | 324(85.12) | 381(69.02) |
| Peripheral Neuropathy | Yes | 56(30.11) | 130(69.89) | 186(33.7) |
| | No | 53(14.48) | 313(85.52) | 366(66.3) |
| Proteinuria | Yes | 44(37.61) | 100(62.39) | 144(26.09) |
| | No | 65(14.94) | 343(85.06) | 408(73.91) |
| SBP | ≤140mmHg | 92(18.59) | 403(81.41) | 495(89.67) |
| | >140mmHg | 17(29.82) | 40(70.18) | 57(10.33) |
| DBP | ≤90mmHg | 96(18.93) | 411(81.07) | 507(91.85) |
| | >90mmHg | 13(28.89) | 32(71.11) | 45(8.15) |
| Pulse BP | ≤60mmHg | 98(18.88) | 421(81.12) | 519(94.02) |
| | >60mmHg | 11(33.33) | 22(66.67) | 33(5.98) |
| HDL | ≤40mg/dl | 31(25.41) | 91(74.59) | 122(22.10) |
| | 40≤60mg/dl | 54(17.48) | 255(82.52) | 309(55.98) |
| | >60mg/dl | 24(19.83) | 97(80.17) | 121(21.92) |
| LDL | ≤130mg/dl | 99(20.84) | 376(79.16) | 475(86.05) |
| | >130mg/dl | 10(12.99) | 67(87.01) | 77(13.95) |
| TC | ≤200mg/dl | 39(19.31) | 163(86.69) | 202(36.59) |
| | >200mg/dl | 70(20.00) | 280(80.00) | 350(63.41) |
| TG | ≤150mg/dl | 16(19.48) | 62(80.52) | 78(14.13) |
| | >150mg/dl | 93(19.79) | 381(80.21) | 474(85.87) |
| FBS | ≤140mg/dl | 9(5.77) | 147(94.23) | 156(28.26) |
| | >140mg/dl | 100(25.25) | 296(74.75) | 396(71.74) |
| HgbA1C | ≤7% | 20(19.42) | 103(80.58) | 103(18.66) |
| | >7% | 89(19.82) | 340(80.18) | 449(81.34) |
| Cr | ≤1.2mg/dl | 86(18.49) | 379(81.51) | 465(84.24) |
| | >1.2mg/dl | 23(26.44) | 64(73.56) | 87(15.76) |

FBS, Fasting Blood Sugar, HDL: High-Density Lipoprotein, HDL: Low-Density Lipoprotein, TC: Total Cholesterol, TG: Triglycerides, HgbA1C: Hemoglobin A1C, Cr: Creatinine

PAD was 29 per 1000 person-years of observation (95% CI: 24–36). At the end of the follow-up period, 9 (1.63%) died, 20 (3.26%) were lost to follow-up, 109 (19.75%) developed PAD, and 47 (8.5%) were transferred to another health institution.

Median time to develop PAD after the diagnosis of T2DM was 13.5 years (95% CI: 11, 14.5). The cumulative survival probabilities of PAD among T2DM patients after 5, 10, and 15 years were 0.97, 0.76, and 0.35, respectively. The overall Kaplan-Meier survivor function estimate shows that events have been proportionally distributed with time and decline steadily. It also shows that the median time to PAD is 13.5 years, and most of the patients develop PAD after 10 years of Type 2 DM diagnosis. The survival of DM patients with PAD decreases progressively (**Fig. 1**).

The survival probability of patients has revealed a relative gap between the categories of certain variables, such as age (**Fig. 2**), sex (**Fig. 3**) and FBS (**Fig. 4**).

The Cox Snell residual plot showed the goodness of fitness of the model was satisfied because the cumulative hazard plot follows 45 degrees (**Fig. 5**).

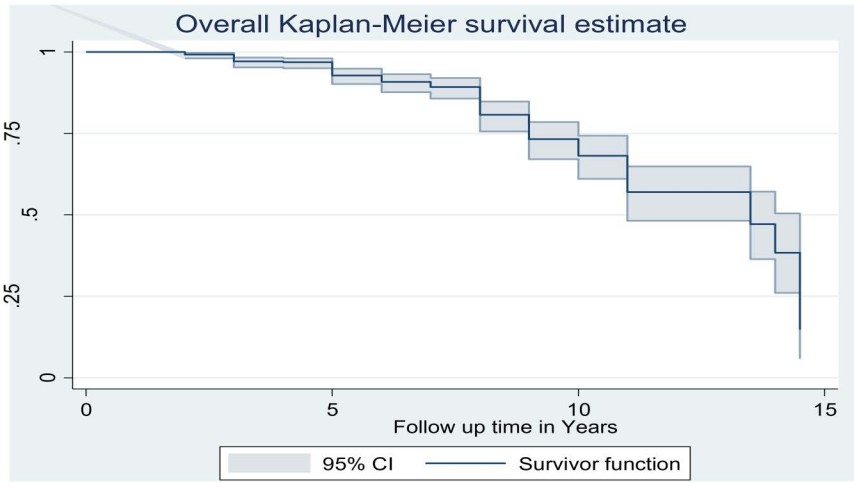

**Fig 1. Overall Kaplan Meir survival curve comparing survival time of adult type 2 DM patients with 95% CI at Felege Hiwot Comprehensive Specialized Hospital, Northwest, Ethiopia, 2023.**

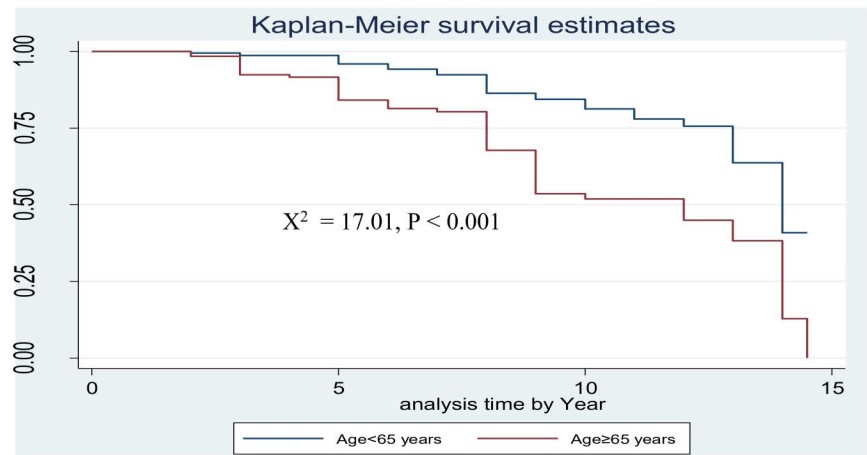

**Fig 2. Kaplan-Meier survival curves to compare PAD among type 2 DM patients between age ≤ 65 years and age > 65 years at FHCSH, Northwest Ethiopia, 2023.**

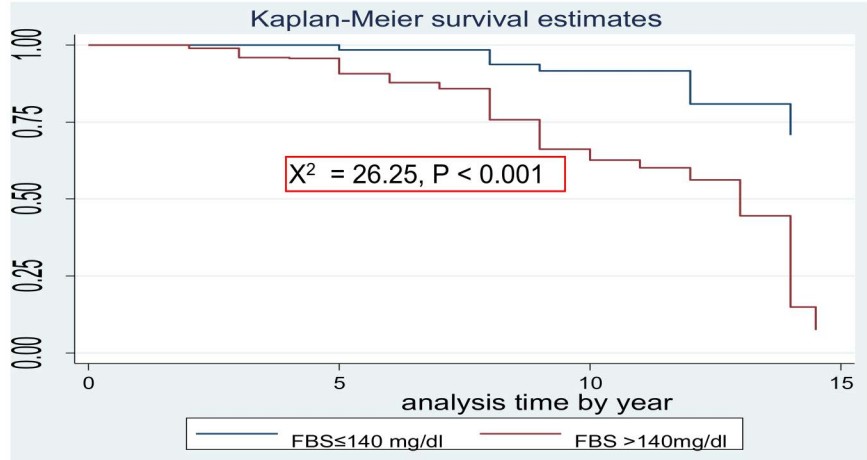

**Fig 3. Kaplan-Meier survival curves to compare PAD among type 2 DM patients between male and female at FHCSH, Northwest Ethiopia, 2023.**

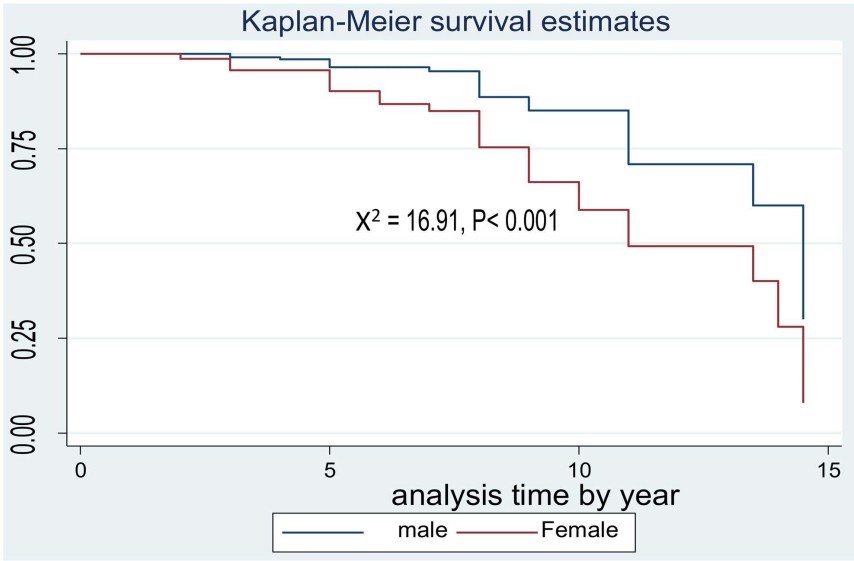

**Fig 4. Kaplan-Meier survival curves to compare PAD among type 2 DM patients between FBS ≤ 140 mg/dl and FBS > 140 mg/dl at FHCSH, Northwest Ethiopia, 2023.**

## Predictors of PAD in adult type 2 DM patients

The bi-variable Cox proportional regression analysis revealed that age, sex, history of hypertension, peripheral neuropathy, triglyceride levels, total cholesterol levels, HbA1C levels, FBS levels, pulse pressure, HDL levels, and family history of hypertension all had p-values below 0.25, qualifying them as potential variables for inclusion in the multivariable Cox proportional hazard analysis.

In the final multivariable Cox proportional regression analysis, age, sex, and FBS were identified as predictors of the development of PAD. The hazard of PAD in type 2 DM patients over the age of 65 was 1.7 times (AHR = 1.66, 95% CI: 1.06, 2.61) higher as compared to

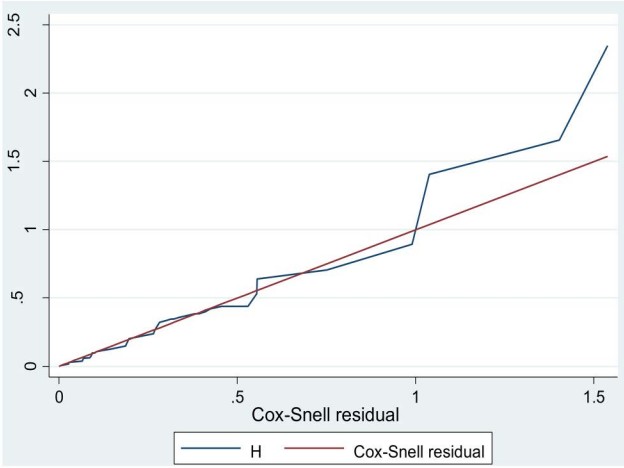

**Fig 5. Cox-Snell residual graph, based on the Kaplan–Meier estimated survivor function, to test the overall adequacy of the cox proportional hazard model of time to PAD and its predictors of type 2 DM patients at FHCSH, 2023.**

those whose age was ≤ 65 years. Moreover, the hazard of PAD in female type 2 DM patients was twofold (AHR = 2.18, 95% CI: 1.36, 3.51), more than in males. Furthermore, the hazard of PAD among type 2 DM patients with baseline FBS > 140 mg/dl was 3.3 times higher than those with FBS ≤ 140 mg/dl (AHR = 3.34, 95% CI: 1.62, 6.90) (**Table 3**).

## Discussion

In this study, the median time to development of PAD was 13.5 years (95% CI: 11, 14.5). However, a study conducted in New Zealand reported a considerably shorter median time of 3 years. The discrepancy may be attributed, in part, to their shorter median follow-up of 2.5 years [14]. Conversely, a study in Ecuador reported a median time to PAD of 26.97 years. This longer time frame may be due to superior patient care receiving their study population, potentially delaying the onset of PAD [15].

In this study, female type 2 DM patients are at a 2-fold increased risk of PAD as compared to males. This finding is in line with a study done in India, which has shown a 2-fold increased risk of female patients developing PAD as compared to male counterparts [22]. These two findings share the same effect as those found in a study done at the University of Gondar, which put female patients at higher risk of PAD [23]. This effect of gender difference could be because of hormonal differences, as sex hormones in females are fluctuant. The decrease in estrogen as women age, which diminishes vasodilators and anti-inflammatory activity, will lead to endothelial dysfunction and, consequently, atherosclerotic arteriosclerosis. Another reason could also be because of sex factors like polycystic ovarian syndrome, preeclampsia, and gestational diabetes mellitus, all of which are associated with vascular endothelial dysfunction [24]. The reason that women are less likely to be engaged in outdoor activities and physical exercise could also explain this sex difference [25].

This study revealed that aging puts type 2 DM patients at high risk of PAD. Patients with an age above 65 are 1.7 times more risky as compared to those below this age. European Prospective Cardiovascular Monitor (PROCAM) has also shown that aging puts these patients at twice the risk of PAD [26]. Reason that aging puts patients at risk of PAD could mainly be because of arteriosclerosis of the arterial wall and stiffness of the arterial muscle due to the

**Table 3. Bi-variable and multivariable Cox regression analysis of adult type 2 DM at Felege Hiwot Comprehensive Specialized Hospital, Northwest Ethiopia, 2023 (n = 552).**

| Variable | Category of Variables | Outcome status | | CHR (95% CI) | AHR (95% CI) | P-value |
|---|---|---|---|---|---|---|
| | | PAD | Censored | | | |
| Sex | Female | 84 | 229 | 2.36(1.51,3.69) | **2.18(1.36,3.51)** * | **0.001** |
| | Male | 25 | 214 | 1 | 1 | |
| Age | ≤65 | 45 | 373 | 1 | 1 | |
| | >65 | 64 | 70 | 2.69(1.82,3.97) | **1.66(1.06,2.61)** * | **0.027** |
| Hypertension | Yes | 43 | 68 | 1.95(1.32,2.88) | 1.35(0.87,2.11) | 0.179 |
| | No | 66 | 375 | 1 | 1 | |
| Pulse pressure | ≤60mmHg | 98 | 421 | 1.45(0.78,2.72) | 1.46(0.70,3.05) | 0.317 |
| | <60mmHg | 11 | 22 | 1 | 1 | |
| Peripheral Neuropathy | Yes | 56 | 130 | 1.31(0.90,1.93) | 1.05(0.67,1.66) | 0.822 |
| | No | 53 | 313 | 1 | 1 | |
| Family history of Hypertension | Yes | 59 | 203 | 1.26(0.86,1.83) | 1.21(0.81,1.83) | 0.355 |
| | No | 50 | 240 | 1 | 1 | |
| HDL | ≤40mg/dl | 31 | 91 | 1 | 1 | |
| | 40≤60mg/dl | 54 | 255 | 0.74(0.47,1.15) | 0.68(0.42,1.07) | |
| | >60mg/dl | 24 | 97 | 0.71(0.41,1.20) | 0.89(0.51,1.57) | |
| TG | ≤150mg/dl | 15 | 62 | 1 | 1 | |
| | >150mg/dl | 94 | 381 | 1.41(0.81,2.44) | 1.32(0.73,2.39) | 0.354 |
| FBS | ≤140mg/dl | 9 | 147 | 1 | 1 | |
| | >140mg/dl | 100 | 296 | 4.60(2.33,9.11) | **3.34(1.62,6.90)** * | **0.001** |
| TC | ≤200mg/dl | 39 | 163 | 1 | 1 | |
| | >200mg/dl | 70 | 280 | 1.30(0.87,1.93) | 1.13(0.73,1.77) | 0.575 |
| HgbA1C | ≤7% | 20 | 103 | 1 | 1 | |
| | >7% | 89 | 340 | 1.15(0.94,1.42) | 1.03(0.62,1.72) | 0.900 |

*Statistically Significant at multivariable with 5% level of significance, CI,Confidence Interval, FBS, Fasting Blood Sugar, HDL, High-Density Lipoprotein, PAD, Peripheral Arterial Disease, TC, Total Cholesterol, TG Triglycerides.

aging process itself. The accumulated effect of endothelial damage and repeated inflammation may cause patients to have arterial stiffness and atheroma in their intima. The decreased arterial flexibility from stiffness and blood flow obstruction by atheroma plaque is what makes the patient be declared to have PAD [27].

This study revealed the hazard of PAD among type 2 DM patients with a baseline FBS > 140 mg/dl to be three times higher than those with ≤ 140 mg/dl. A study done in Taiwan has also shown those patients with high FBS to be at higher risk of PAD [28]. This could be because of the effect of hyperglycemia on top of the characteristic insulin resistance of type 2 DM patients, which results in metabolic abnormalities. These metabolic abnormalities cause an overproduction of reactive oxygen species, which have a crucial role in precipitating diabetic vascular diseases via inflammation and endothelial dysfunction [29].

This study revealed that the incidence rate of PAD was 29 per 1,000 person-years of observation (95% CI: 24–36) during the follow-up period. This finding is similar to the incidence of PAD observed in type 2 diabetes patients in a study conducted in the USA [6]. However, this rate is higher compared to a study conducted in India, which reported an incidence rate of 7.6 [5], and Ecuador, which reported an incidence rate of 23.38 per 1,000 person years of observation [15]. The higher incidence of PAD observed in our study compared to others may

be attributed to difference in case detection methods. While previous studies primarily used on the ankle-brachial index (ABI < 0.9), which can underestimate PAD incidence, our study employed frequent ultrasound evaluations. This more sensitive approach likely identified asymptomatic cases, resulting a higher diagnostic rate. Additionally, variations in study populations, such as differences in self- management practices and treatment adherence among Indian and Ecuadorian participants, may also contribute the observed disparity. Furthermore, social and cultural and governmental interventions, along with differences in sample size and study design, could further explain these disparities.

## Public health and research implications

This study offers critical insights for clinicians, researchers, and policymakers seeking to reduce PAD related morbidity and mortality in patients with diabetes mellitus. The high PAD incidence highlights the urgent need for comprehensive, patient-centered services. The predictors identified will enable healthcare professionals to better anticipate PAD risk and proactively manage the disease. Furthermore, our findings will inform the development of cost-effective strategies for early diagnosis, prevention, and treatment of diabetes and its complications. Finally, the study will serve as a foundation for future research aimed at further understanding and controlling PAD in type 2 diabetes.

## Strengths and limitations of this study

To our knowledge, this is the first study to investigate the time to detection of PAD incidence rates and predictors in Ethiopia patients with type 2 diabetes mellitus. While this study provides novel insights, it is important to acknowledge its limitations. Incomplete data recording prevents as from examining the influence of certain socio-demographic and behavioral factors, such as education level, alcohol consumption, smoking status, income, exercise, and body mass index. Additionally, while Doppler ultrasound was used to diagnose PAD in most participants, operator-dependent nature could introduce variability in interpretations. It's important to note that while Doppler ultrasound is helpful in assessing the vascular complications of diabetes, it is not a standard diagnostic tool for diabetes itself and should be used in conjunction with other diagnostic methods and clinical evaluations.

## Conclusion and recommendation

Our study reveals that most PAD cases in type 2 DM patients occur more than 10 years after their diabetes diagnosis, with a high overall incidence rate and a cumulative survival probability of 34.91% for patients remaining PAD-free after the final year of follow-up. We identified female sex, older age, and elevated baseline FBS as significant predictors of PAD development in this population. These findings underscore the urgent need for effective preventive and control strategies for diabetes complications in Ethiopia. We recommend that healthcare professionals prioritize frequent PAD screening for women and elderly diabetic and strongly emphasize strict blood sugar control. Further prospective studies should investigate the role of additional factors such as body mass index, smoking, and alcohol consumption, in PAD development.

## Acknowledgment

The authors are very grateful to Bahir Dar University, College of Medicine and Health Science, School of Public Health, and FHCSH Quality and Research Unit for their permission to conduct the study. We are sincerely grateful to all staff working at the medical referral clinic as well as the recording and reception unit at FHCSH. We would also like to extend our heartfelt thanks to data collectors and supervisors.

## Author contributions

**Conceptualization:** Dessalew Abelneh Woleli, Gebiyaw Wudie Tsegaye, Taye Abuhay.

**Data curation:** Dessalew Abelneh Woleli, Gebiyaw Wudie Tsegaye, Taye Abuhay, Abyot Terefe Teshome, Gebrie Getu Alemu.

**Formal analysis:** Dessalew Abelneh Woleli, Gebiyaw Wudie Tsegaye, Taye Abuhay, Abyot Terefe Teshome, Gebrie Getu Alemu.

**Funding acquisition:** Dessalew Abelneh Woleli, Gebiyaw Wudie Tsegaye, Taye Abuhay, Abyot Terefe Teshome, Gebrie Getu Alemu.

**Investigation:** Dessalew Abelneh Woleli, Gebiyaw Wudie Tsegaye, Taye Abuhay, Abyot Terefe Teshome, Gebrie Getu Alemu.

**Methodology:** Dessalew Abelneh Woleli, Gebiyaw Wudie Tsegaye, Taye Abuhay, Abyot Terefe Teshome, Gebrie Getu Alemu.

**Project administration:** Dessalew Abelneh Woleli, Gebiyaw Wudie Tsegaye, Taye Abuhay, Abyot Terefe Teshome, Gebrie Getu Alemu.

**Resources:** Dessalew Abelneh Woleli, Gebiyaw Wudie Tsegaye, Taye Abuhay, Abyot Terefe Teshome, Gebrie Getu Alemu.

**Software:** Dessalew Abelneh Woleli, Gebiyaw Wudie Tsegaye, Taye Abuhay, Abyot Terefe Teshome, Gebrie Getu Alemu.

**Supervision:** Dessalew Abelneh Woleli, Gebiyaw Wudie Tsegaye, Taye Abuhay, Abyot Terefe Teshome, Gebrie Getu Alemu.

**Validation:** Dessalew Abelneh Woleli, Gebiyaw Wudie Tsegaye, Taye Abuhay, Gebrie Getu Alemu.

**Visualization:** Dessalew Abelneh Woleli, Gebiyaw Wudie Tsegaye, Taye Abuhay, Abyot Terefe Teshome, Gebrie Getu Alemu.

**Writing – original draft:** Dessalew Abelneh Woleli, Gebiyaw Wudie Tsegaye, Taye Abuhay.

**Writing – review & editing:** Abyot Terefe Teshome, Gebrie Getu Alemu.

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
