## [Decision Letter · Decision Letter 0]

26 Jun 2024

PONE-D-23-41579The incidence and predictors of peripheral arterial disease among type 2 diabetes mellitus patients at Felege Hiwot Comprehensive Specialized Hospital, Northwest Ethiopia, 2023: a retrospective follow up studyPLOS ONE

Dear Dr. Alemu,

Thank you for submitting your manuscript to PLOS ONE. After careful consideration, we feel that it has merit but does not fully meet PLOS ONE’s publication criteria as it currently stands. Therefore, we invite you to submit a revised version of the manuscript that addresses the points raised during the review process.

We look forward to receiving your revised manuscript.

Kind regards,

Wondimeneh Shiferaw

Academic Editor

PLOS ONE

Journal Requirements:

Reviewers' comments:

Reviewer's Responses to Questions

**Comments to the Author**

1. Is the manuscript technically sound, and do the data support the conclusions?

Reviewer #1: Partly

Reviewer #2: Yes

2. Has the statistical analysis been performed appropriately and rigorously? 

Reviewer #1: Yes

Reviewer #2: Yes

3. Have the authors made all data underlying the findings in their manuscript fully available?

Reviewer #1: Yes

Reviewer #2: Yes

4. Is the manuscript presented in an intelligible fashion and written in standard English?

Reviewer #1: No

Reviewer #2: No

5. Review Comments to the Author

Reviewer #1: 1. There are linguistic and editing errors like lack of hyphens, periods, inconsistent capitalization, misspellings, and formatting inconsistencies.

2. Can you provide more specific details on the existing research gap in Ethiopia regarding PAD and vascular complications in type 2 diabetes mellitus patients? How does the proposed study aim to address this gap, and what unique contributions does it offer to the existing literature?

3. You focused your study on type 2 diabetic patients aged 14 years and older. Could you explain why you didn't include children under 14, considering that type 2 diabetes is also affecting younger individuals? Additionally, what's the rationale behind choosing this particular age as the cutoff point?

4. It would be better if the discussion section discusses potential differences in study populations, methodologies, and healthcare systems that may contribute to variations in findings. Additionally, considering factors such as sample size and follow-up duration could provide further context for interpretation.

5. The discussion addresses variations in PAD incidence rates across different studies and attributes the higher incidence rate in this study to differences in case detection methodologies. However, it would be helpful to discuss potential limitations of Doppler ultrasound evaluations in detecting PAD, such as operator dependence and variability in interpretation, and whether alternative diagnostic approaches could provide complementary information.

6. Regarding the predictors identified for PAD development in type 2 DM patients, how could healthcare professionals integrate this information into clinical practice to improve patient care and outcomes?

7. Are there any unique challenges or opportunities in Ethiopia's healthcare system or socio-economic environment that should be considered when interpreting the study findings and implementing interventions?

Reviewer #2: Review report

Manuscript tittle: The incidence and predictors of peripheral arterial disease among type 2 diabetes mellitus patients at Felege Hiwot Comprehensive Specialized Hospital, Northwest Ethiopia, 2023: a retrospective follow up study Short Title: Incidence and predictors of peripheral arterial disease among type 2 diabetes mellitus patients

Manuscript Number: PONE-D-23-41579

In the abstract line 40 Result: please write as Results

Being female (AHR=2.18(1.36, 3.51), age 42 above 65 years (AHR=1.66, CI: 1.06, 2.61), and fasting blood sugar of more than 140mg/dl 43 (AHR= 3.34(1.62, 6.90) were predictors for time to peripheral arterial disease in type 2 diabetes 44 mellitus.( Line 42- 44)

Comments: AHR: Please write the entire word (definition) first, followed by its abbreviation in brackets, and then utilize it throughout your write up. Please make the same corrections for the remaining abbreviations in this document.

Please add 95% CI consistently

In the introduction section (line 77)

Some studies have estimated the median times for onset of PAD in type2 diabetes mellitus patients

Comments: type2 correct it as type 2

There are factors which can affect the survival of type 2 DM patients from PAD occurrence. (Line 79)

Comments: DM: Please write the entire word (definition) first, followed by its abbreviation in brackets, and then utilize it throughout your write up. Please make the same corrections for the remaining abbreviations in this document.

The material and methods is Ok, just to delete some redundancy sentences

Comments: Please delete the followings in the material and methods section.

Study design (line 99)

An institution-based retrospective follow-up study was conducted at Felege Hiwot Comprehensive Specialized Hospital, Northwest Ethiopia (line 100-101)

Please rewrite as Study design, area and period in line 102

Please add Independent variables as a subtitle after line 153

Results

The cumulative survival to PAD probabilities of T2DM patients after 5, 10 and 15 years were 0.97, 0.76, 0.35 respectively.(Lines 261-262)

Comments: please rewrite the above sentence as The cumulative survival probabilities of PAD among T2DM patients after 5, 10 and 15 years were 0.97, 0.76 and 0.35, respectively.

Please rewrite the sentence in line 280 as (AHR=1.66, 95% CI: 1.06, 2.61) higher as compared to those whose age was ≤65 years.

Discussion

The importance or implication or significance of each result should be stated in the discussion section.

In the Conclusion section do not use figures (numbers) rather conclude the main findings.

Figures 1- 4 needs edition for example analysis time by what? month or year? Clearly state it.

Chi2 please replace it by X2

6. PLOS authors have the option to publish the peer review history of their article (what does this mean? ). If published, this will include your full peer review and any attached files.

**Do you want your identity to be public for this peer review?** For information about this choice, including consent withdrawal, please see our Privacy Policy .

Reviewer #1: No

Reviewer #2: **Yes: ** Tamiru Alene

---

## [Author Response · Author response to Decision Letter 1]

5 Aug 2024

To: PLOS ONE Editorial Office

Authors’ point-by-point Responses to a Manuscript ID: PONE-D-23-41579

Dear Editor in Chief, Greetings.

We have submitted these authors' responses to peer-review comments and questions for a manuscript entitled: The incidence and predictors of peripheral arterial disease among type 2 diabetes mellitus patients at Felege Hiwot Comprehensive Specialized Hospital, Northwest Ethiopia, 2023: a retrospective follow-up study.

Dear Editor, first, we would like to thank the PLOS ONE Editorial Office members, especially the chief editor, who timely assigned the competent academic editor and facilitated the progress of the review forum. Second, our special thanks go to the academic editor for his editorial contribution and for assigning the potential skilled and experienced reviewers in the field promptly. Moreover, our deepest gratitude goes to the esteemed reviewers for their constructive comments and scientific contributions that helped us improve the quality of this manuscript.

Dear academic Editor and reviewers, we have made the necessary corrections and responses to those comments raised by reviewers 1 and 2 point by point, page by page, and line by line sequentially. Accordingly, we have added the authors’ responses in the review forum, as well as the revised manuscript and the tracked changes to the manuscript on the web page.

With regards!

Gebrie Getu Alemu

gebryegetu27@gmail.com

University of Gondar, Ethiopia

PO. Box 196

Corresponding Author

---

## [Decision Letter · Decision Letter 1]

29 Jan 2025

PONE-D-23-41579R1The incidence and predictors of peripheral arterial disease among type 2 diabetes mellitus patients at Felege Hiwot Comprehensive Specialized Hospital, Northwest Ethiopia, 2023: a retrospective follow up studyPLOS ONE

Dear Dr. Alemu,

Thank you for submitting your manuscript to PLOS ONE. After careful consideration, we feel that it has merit but does not fully meet PLOS ONE’s publication criteria as it currently stands. Therefore, we invite you to submit a revised version of the manuscript that addresses the points raised during the review process.

The manuscript has been evaluated by two reviewers, and their comments are available below.

The reviewers have a couple of minor clarification requests. Could you please revise the manuscript to carefully address the concerns raised?

We look forward to receiving your revised manuscript.

Kind regards,

Helen Howard

Staff Editor

PLOS ONE

Journal Requirements:

Reviewers' comments:

Reviewer's Responses to Questions

**Comments to the Author**

1. If the authors have adequately addressed your comments raised in a previous round of review and you feel that this manuscript is now acceptable for publication, you may indicate that here to bypass the “Comments to the Author” section, enter your conflict of interest statement in the “Confidential to Editor” section, and submit your "Accept" recommendation.

Reviewer #1: All comments have been addressed

Reviewer #3: All comments have been addressed

2. Is the manuscript technically sound, and do the data support the conclusions?

Reviewer #1: Yes

Reviewer #3: Yes

3. Has the statistical analysis been performed appropriately and rigorously? 

Reviewer #1: Yes

Reviewer #3: Yes

4. Have the authors made all data underlying the findings in their manuscript fully available?

Reviewer #1: Yes

Reviewer #3: Yes

5. Is the manuscript presented in an intelligible fashion and written in standard English?

Reviewer #1: Yes

Reviewer #3: Yes

6. Review Comments to the Author

Reviewer #1: Thank you for your revised submission and for carefully addressing the concerns I raised. I appreciate the efforts you have made to improve the manuscript. The revisions have significantly strengthened the clarity and overall quality of the paper.

Reviewer #3: In the abstract – Result sub-section:

“The incidence rate of peripheral arterial disease was high.”

The authors need to specify the reference to conclude high like compare with the recommended value or compare with the previous study or -----

In the result section

The hazard of PAD in female type 2 DM patients was twofold (AHR = 2.18), more

283 than in males. Furthermore, the hazard of PAD among type 2 DM patients with baseline FBS >284 140 mg/dl was 3.3 times higher than those with FBS ≤ 140 mg/dl (AHR = 3.34).

The authors need to include 95%CI.

7. PLOS authors have the option to publish the peer review history of their article (what does this mean? ). If published, this will include your full peer review and any attached files.

**Do you want your identity to be public for this peer review?** For information about this choice, including consent withdrawal, please see our Privacy Policy .

Reviewer #1: No

Reviewer #3: **Yes: ** Esubalew Tesfahun

---

## [Author Response · Author response to Decision Letter 2]

31 Jan 2025

To: PLOS ONE Editorial Office

Authors’ point-by-point Responses to a Manuscript ID: PONE-D-23-41579R1

Dear Editor in Chief, Greetings.

We have submitted these authors' responses to peer-review comments and questions for a manuscript entitled: The incidence and predictors of peripheral arterial disease among type 2 diabetes mellitus patients at Felege Hiwot Comprehensive Specialized Hospital, Northwest Ethiopia, 2023: a retrospective follow-up study.

Dear Editor, first, we would like to thank the PLOS ONE Editorial Office members, especially the chief editor, who timely assigned the competent academic editor and facilitated the progress of the review forum. Second, our special thanks go to the academic editor for his editorial contribution and for assigning the potential skilled and experienced reviewers in the field promptly. Moreover, our deepest gratitude goes to the esteemed reviewers for their constructive comments and scientific contributions that helped us improve the quality of this manuscript.

Dear academic Editor and reviewers, we have made the necessary corrections and responses to those comments raised by reviewers. Accordingly, we have added the authors’ responses in the review forum, as well as the revised manuscript and the tracked changes to the manuscript on the web page.

With regards!

Gebrie Getu Alemu

gebryegetu27@gmail.com

University of Gondar, Ethiopia

PO. Box 196

Corresponding Author

---

## [Editor Report · Decision Letter 2]

10 Feb 2025

PONE-D-23-41579R2

The incidence and predictors of peripheral arterial disease among type 2 diabetes mellitus patients at Felege Hiwot Comprehensive Specialized Hospital, Northwest Ethiopia, 2023: a retrospective follow up study

PLOS ONE

Dear Dr. Alemu,

Thank you for submitting your manuscript to PLOS ONE. After careful consideration, we feel that it has merit but does not fully meet PLOS ONE’s publication criteria as it currently stands. Therefore, we invite you to submit a revised version of the manuscript that addresses the points raised during the review process.

Specifically, please note that PLOS ONE does not copyedit manuscripts, so the language in submitted articles must be clear, correct, and unambiguous. Please work to improve the quality of the writing throughout your manuscript. We recommend enlisting the help of a professional copyediting service.

We look forward to receiving your revised manuscript.

Kind regards,

Helen Howard

Staff Editor

PLOS ONE

Journal Requirements:

Additional Editor Comments:

- Please work to improve the quality of the writing throughout your manuscript. We recommend enlisting the help of a professional copyediting service.

---

## [Author Response · Author response to Decision Letter 3]

12 Feb 2025

Dear Editor in Chief, Greetings.

We have submitted these authors' responses to editor’s comments and questions for a manuscript entitled: The incidence and predictors of peripheral arterial disease among type 2 diabetes mellitus patients at Felege Hiwot Comprehensive Specialized Hospital, Northwest Ethiopia, 2023: a retrospective follow-up study.

Dear Editor, first, we would like to thank the PLOS ONE Editorial Office members, especially the chief editor, who timely assigned the competent academic editor and facilitated the progress of the review forum. Second, our special thanks go to the academic editor for his editorial contribution and for assigning the potential skilled and experienced reviewers in the field promptly.

Dear academic Editor we have made the necessary corrections and responses to those comments raised by editors. Accordingly, we have added the authors’ responses in the review forum, as well as the revised manuscript and the tracked changes to the manuscript on the web page.

With regards!

Gebrie Getu Alemu

gebryegetu27@gmail.com

University of Gondar, Ethiopia

PO. Box 196

Corresponding Author

---

## [Editor Report · Decision Letter 3]

16 Feb 2025

PONE-D-23-41579R3The incidence and predictors of peripheral arterial disease among type 2 diabetes mellitus patients at Felege Hiwot Comprehensive Specialized Hospital, Northwest Ethiopia, 2023: a retrospective follow up studyPLOS ONE

Dear Dr. Alemu,

Thank you for submitting your manuscript to PLOS ONE. After careful consideration, we feel that it has merit but does not fully meet PLOS ONE’s publication criteria as it currently stands. Therefore, we invite you to submit a revised version of the manuscript that addresses the points raised during the review process.

We understand that the Grammarly tool was used to assist you with this revision, but unfortunately language errors remain throughout the manuscript. The manuscript still requires copyediting in order to improve the English grammar and syntax to a publishable standard. We recommend asking a native English-speaking colleague to assist you or to enlist the help of a professional copyediting service.

We look forward to receiving your revised manuscript.

Kind regards,

Helen Howard

Staff Editor

PLOS ONE
---

## [Author Response · Author response to Decision Letter 4]

22 Feb 2025

We have submitted these authors' responses to editor’s comments for a manuscript entitled: The incidence and predictors of peripheral arterial disease among type 2 diabetes mellitus patients at Felege Hiwot Comprehensive Specialized Hospital, Northwest Ethiopia, 2023: a retrospective follow-up study.

Dear Editor, first, we would like to thank the PLOS ONE Editorial Office members, especially the chief editor, who timely assigned the competent academic editor and facilitated the progress of the review forum. Second, our special thanks go to the academic editor for his editorial contribution and for assigning the potential skilled and experienced reviewers in the field promptly.

Dear academic Editor we have made the necessary corrections and responses to those comments raised by editors. Accordingly, we have added the authors’ responses in the review forum, as well as the revised manuscript and the tracked changes to the manuscript on the web page.

---

## [Editor Report · Decision Letter 4]

27 Feb 2025

The incidence and predictors of peripheral arterial disease among type 2 diabetes mellitus patients at Felege Hiwot Comprehensive Specialized Hospital, Northwest Ethiopia, 2023: a retrospective follow up study

PONE-D-23-41579R4

Dear Dr. Alemu,

We’re pleased to inform you that your manuscript has been judged scientifically suitable for publication and will be formally accepted for publication once it meets all outstanding technical requirements.

Kind regards,

Patrick Goymer

Staff Editor

PLOS ONE
---

## [Editor Report · Acceptance letter]

PONE-D-23-41579R4

PLOS ONE

Dear Dr. Alemu,

I'm pleased to inform you that your manuscript has been deemed suitable for publication in PLOS ONE. Congratulations! Your manuscript is now being handed over to our production team.

Kind regards,

on behalf of

Dr Patrick Goymer

Staff Editor

PLOS ONE